# Degradation of Mechanical Properties of Pine Wood Under Symmetric Axial Cyclic Loading Parallel to Grain

**DOI:** 10.3390/polym12102176

**Published:** 2020-09-23

**Authors:** Mariana D. Stanciu, Horațiu D. Teodorescu, Sorin Vlase

**Affiliations:** Department of Mechanical Engineering, Transilvania University of Brașov, B-dul Eroilor 29, 500036 Brașov, Romania; draghicescu.teodorescu@unitbv.ro (H.D.T.); svlase@unitbv.ro (S.V.)

**Keywords:** pine wood, symmetric axial cyclic loading, viscous-elastic behavior, degradation of mechanical properties, creep

## Abstract

The mechanical properties of wood, respectively the elastic, plastic, and strength properties, depend on a large number of factors, due both to its structural and physical characteristics, as well as to the size, direction, nature, and speed of application of forces. Wood, generally considered to be a viscous-elastic material, has creep deformations over time under the effect of a constant load. In this study the behavior of pine wood samples was investigated due to its large utilization in different finished products, such as roof construction, furniture, outdoor applications, garden furniture, and toys. The paper aims to analyze the viscoelastic behavior of pine wood subjected to cyclically loading to traction-compression with different loads (1 kN; 1.5 kN; 2 kN), applied at different speeds (1 mm/min; 10 mm/min). It was observed that, at low speeds (1 mm/min) and low intensities of the applied force, it was possible to distinguish the three creep regions specific to wood: the primary area (primary flow), the secondary area, and finally the tertiary creep. As the force increases, the law of variation of the wood flow changes. The degradation of longitudinal elasticity modulus occurs with the increase of the number of cycles, so after 20 alternating symmetrical cycles of traction-compression of the pine wood samples, there is a decrease of its values by 35%.

## 1. Introduction

Wood, due to its polymeric and layered structure, is considered a natural composite material. Thus, the late wood and early wood can be considered composite layers [1,2]. Their widths, proportion and density vary, depending on the specific environment and soil conditions from year to year [3]. Wood consists of a very complex system of fibers, cells, cementing substances, etc., whose physical and chemical cohesion poses particular problems in the calculations of strength and rigidity, compared to homogeneous and isotropic materials such as, for example, metals [4,5,6]. Nowadays, wood and wood-based materials have re-entered the attention of architects and builders, being a material with numerous physical, mechanical, acoustic, and workability qualities. Numerous studies on the mechanical properties of softwood used in construction have been conducted to highlight the correlations between wood structure, quality class, type of mechanical stress, and failure modes [2,7,8].

Taking into account the structural changes produced by cyclic stresses as well as the modification of the tensile breaking section of wooden samples, Christoforo et al. [4] present a mathematical model for determining the shape coefficient that expresses the relationship between the actual breaking area and the theoretical area. The tension is distributed at the request of parallel traction with fiber of the different wood species from Brazil. 

However, the cyclic stresses of wooden structures lead to a constant reduction of wood strength by accumulating deformation energy and producing plastic deformations at the cell walls. Studies have shown that, to predict the behavior of wood over time, the design and construction of structural elements must take into account not only the values of allowable stresses, but also the ability to load wood under cyclic load, respectively the creep and relaxation function of wood [9,10,11]. Gong et al. [1] analyzed the failure mechanism of softwood subject to compression cyclic stresses parallel to the fibres, noting that the process of mechanical degradation is accompanied by a structural degradation characterized by three stages: kink initiation, kink growth, and finally macroscopic failure. Similar to the composite materials and kinking fibres of them studied by Pinho et al. [12], the wood fibres kinking failure occurs because of local microstructural defects, such as fibres misalignments and longitudinal cracking. the wood fibres misalignments lead to shearing stresses between the fibres and longitudinal cracks or parenchyma rays’ disposals act as stress concentrators.

Konstantinov et al. [9] analyzed the influence of the texture orientation of the wood on the strength and deformation properties of wood and also the degradation of mechanical and dynamical properties of wood subjected to cyclic loads in the conditions of lateral confinement, being studied three species: pine, birch and sequoia. They noticed that at a low strain rate, the damage to the specimen was seen to be insignificant, while at a high strain rate the integrity of the specimen was exceeded leading to sample failure. Yildirim et al. [13] examined the fatigue and static strength of Scots pine and beech wood subjected to bending test. They noticed that the fatigue failures began to occur as the stress level was or above 40% of Scots pine specimen’s average of modulus of rupture (MOR) and for beech specimens, fatigue failures began to occur when the stress level was above 50% of beech specimen’s average MOR. Some studies analyze the fatigue behavior of chemically or thermally modified wood compared to control samples, finding that both the modulus of bending elasticity (MOE) increased by about 21% in the case of pine wood modified by impregnation with phenol formaldehyde resin [10]. Regarding the behaviour of Scots pine (*Pinus sylvestris*) as control samples and thermally modified subjected to cyclic bending, Sharapov et al. [11] performed studies applying asymmetric sinusoidal cyclic oscillations at 20 Hz frequency and load ratio p = 0.3. Three types of variables they analyzed in experiments: the thermal treatment temperature (160, 190, and 220 °C), the number of bending oscillations (103, 505 × 103 and 106), and the equilibrium moisture content at target climates of 20 °C and 35%, 65%, and 95% relative humidity (RH). Sharapov et al. [11] considered that the main influence on the residual MOR consist of the initial moisture content of the specimens before fatigue testing and the maximum thermal modification temperature. Based on experimental observations and mathematical models, Hassani et al. [14] and Andrianopoulos et al. [15] developed a three-dimensional orthotropic mechanical and mathematical model of wood, taking into account both the components of total deformation and rheological expressions, principles of moisture-stress analysis. In previous studies, Stanciu et al. [16,17] investigated the rheological characteristics of the black locust wood samples chemical treated with ammonia [16] and natural fibers composites [17] in terms of complex modulus, with its two components (the conservation modulus E’ and the loss modulus E”) and the damping factor were determined dynamic mechanical analysis (DMA). 

While numerous studies have presented failure analyses of wood subjected to axial stresses (or tensile or compression) or flexural stresses, little information is available about the structural and mechanical degradation of wood (softwood) subjected to symmetric axial cyclic tension, obtaining the Bauschinger effect. Wooden constructions, new or heritage, are subject to variable loading, such as precipitation (snow), wind, or dynamic loads from earthquakes, which requires a knowledge of the mechanical properties of wooden components, behavior over time, along with a description detailed defects or their degradation (resistance reduction factors). Pine wood, along with spruce wood, and fir wood, is very widely used in making construction/building elements that are subjected to loads in different directions and with different intensities and duration. So, the objective of the current study is to analyze the degradation of mechanical properties and structural failure produces by symmetric axial cyclic loading of Scots pine samples (*Pinus sylvestris*). The novelty of the paper consists of better understanding of the mechanical behavior of softwood and determination of the hysteresis loops to tensile and compression. Also, the structural modifications of wood cells are presented by means of optical microscope.

## 2. Materials and Methods 

### 2.1. Preliminary Tests

The wood species analyzed in this research was Scots pine samples (*Pinus sylvestris*) with 12% moisture contents (provided by the norm EN 384: 2004) [18]. In the first stage, a set of samples with different loading speeds were tested for tensile to rupture, in order to establish the maximum value of the load to alternating symmetrical cyclic stresses (Figure 1a). In Figure 1b, the characteristic curves of pine wood samples to different speed loading are presented. It can be noticed that the yield point for loading speed of 1 mm/min is around of 1kN force and for loading speed of 10 mm/min, yield point corresponds to a load of 2 kN. From the preliminary results, the cyclic loading will range between 1 kN and 2 kN, with loading speeds of 1 mm/min and 10 mm/min.

### 2.2. Materials

Based on ASTM 04.10 Wood D143-09 [19] and EN 408 (2010) + A1 (2012) [20], the procedures and samples for axial cyclic tests were developed. The 18th cases as combination between loading intensity, number of cycles and speed of loading were analyzed on samples of solid wood of Scots pine cut radially, having the nominal dimensions of length *L* = 150 mm, width *b* = 15 mm, thickness *h* = 5 mm (Figure 2a). The Scots pine wood samples used in this study come from the Carpathian Mountains but they are provided by the trade market where ordinary customers buy it. The annual ring width of specimens ranges between 1.5 mm to 3 mm, so the Scots pine wood samples shows larger spreading of its annual values as Silvestru Grigore et al. [3] noticed. 

The first category of tests denoted A1 was the one in which the samples were subjected to loading with a speed of 1 mm/min with the force intensity of ±1000 N, being formed by three sets of samples. The first set of samples P1, P2, P3 was subjected to the tensile-compression stresses parallel to wood grain in turn at 1 cycle (sample P1); at 10 cycles (sample P2) and at 20 cycles (sample P3). The next set of samples (B1) was subjected to symmetrical cyclic stresses with a speed of 1 mm/min with the force intensity of ±1500 N in the same manner as previous samples: at 1 cycle (sample P4); at 10 cycles (sample P5) and at 20 cycles (sample P6). The C1 group of samples was subjected to cyclic stress at a load intensity of ±2000 N, the loading speed of 1 mm/min, at 1 cycle (sample P7), at 10 cycles (sample P8), and at 20 cycles, (sample P9). The second category of pine specimens denoted with A10; B10; C10 was subjected at a speed of 10 mm/min, having similarly to the first category, three sets of samples tested at different intensities of the applied load. The input data of tested samples are presented in Table 1. 

### 2.3. Methods

The samples were subjected to symmetric cyclic loading tensile-compression test using the universal testing machine type LS100 Lloyd’s Instrument belonging to the Mechanical Engineering Department of Transilvania University of Brasov, in order to study the degradation of mechanical properties of a Scots pine wood. The specimens were loading with a constant speed of 1 mm/min and then 10 mm/min, being subjected successive to 1 cycle, 10 cycles and 20 cycles. The tensile and compression displacements were measured simultaneously with applied loading (Figure 2b,c). For data acquisition, the Nexygen Plus software was used. After the tests, the characteristic curve, the specific deformation, the longitudinal elastic modulus, the rupture tension of each samples were determined, and on the basis of the load curves, the average deformation energy for each type of sample was calculated. The fracture of samples was analyzed with optical devices. Figure 2 shows the experimental equipment and some details from tests.

## 3. Results and Discussion 

During the experimental investigations, numerous primary data were obtained that required processing so that the viscous-elastic behavior of pine wood can be characterized. Thus, in Figure 3 are presented the variations of the tensile-compression loading.

Figure 4 shows the hysteresis loop for the first stress cycle, being highlighted the values of lengthening and shortening the sample. The total strain range Δε (the strains from A to C points) is calculated as total of tensile strain and compression strain containing both elastic and plastic strains. The true elastic strain range Δε_e_ (the strains from A to D points) represents the recovery strain after a complete cycle and total true plastic strain range Δε_p_ (the strains from D to 0 points) is the permanent strain after a complete loading cycle. The tensile recovered elastic strain from point B to point 0 can be observed in Figure 4a–f. In Table 2 are summarized the strains values for the first request cycle. Analyzing the six cases presented, it is observed that the lengthenings (traction strain) are greater than the shortenings (compression strain), pine wood being an anisotropic material, with a different behavior when subjected to traction, respectively compression. The difference between extensions and shortenings depends on the intensity of the force applied and the loading speed. As the applied load intensity increases, the percentage difference between traction and compression displacement increases, and as the load speed increases, they increase. Based on the Boltzmann superposition principle, which describes the response of a material to different loading histories and fitting the deformations peaks for each cycle [21,22], in Figure 5a–f are graphically represented the master curves covering the accumulated peak deformations in tensile (Figure 5a,c,e) and compression tests (Figure 5b,d,f). The behavior of the tested specimens differs in the two types of axial stresses. For the force of 1 kN and the stress rate of 1 mm/min, they can be distinguished in the graph in Figure 5a, the three wood-specific flow zones: the primary zone (primary flow) in which the displacements / deformations have a high growth rate from one cycle to another (the first four cycles), the secondary area in which there is a stagnation of displacements / deformations (the next 4–5 cycles), and finally, the tertiary creep in which there is again an increase in the speed of movement / deformation of the wood. This behavior is noticeable at low load speeds and low force intensities. As the force increases, the law of variation of the wood creep changes (Figure 5c,e). In the case of the variation of the compression strains, a rapid increase of them is observed with the increase of the stress cycles, regardless of the intensity of the force, as can be seen in Figure 5b,d,f. The main three zone in case of compression can be difficult distinguished, but the viscous-elastic behavior of wood is more pronounced in these cases (Figure 5c–f). These curves express the speed of accumulation of plastic deformations, which ultimately lead to damage of the structure.

The most important mechanical degradations occur in the first stress cycles as seen on the creep curve, specifically the first area (Figure 5). Thus, at a load of 1 kN, the longitudinal modulus of elasticity degrades by 12% of the initial value, compared to 20 cycles when the degradation is 15% of the initial value. (Figure 6). Increasing the loading intensity leads to increasing the degradation rate of 19% after 20 cycles in case of 1.5 kN and 28% in case of load of 2 kN. Increases the speed loading of 10 mm/min lead to a slight decreases of degradation rate (12%; 18% and 21% for each value of loading intensity). The obtained values are comparable with Ferreira et al. [23] which studied the pine wood samples subjected to tensile stresses. The mean value of elasticity modulus in the longitudinal direction parallel to the grain obtained by Ferreira et al. [23] is 4141 MPa, the sizes of samples being similar to those in the present study. 

The stress values are independent of the number of stress cycles and the stress rate. These have the same values both in traction and compression. The difference between the categories of values is given by the intensity of the applied loading. Thus, for the force of ±1 kN, the average value of the maximum stress ranges between 13 and 14 MPa, for the force of ±1.5 kN, the average value of the maximum stress is between 19 and 20 MPa, and for the force of ±2 k, the average value of the maximum stress is between 26 and 27 MPa, as can be seen in Figure 7.

The values obtained give us an image of the quality class of the Scots pine samples studied, namely that they are part of quality class C45 according to wood standard. As the study was performed on small samples, it can be improved by additional studies on larger length samples to correlate the obtained results with the inhomogeneity and anisotropy of the material. 

In Figure 8, the percentage variations of tensile and compression strains are presented for each studied case. The compression strains represent approximately 40% of the total deformation of the samples, the tensile strains being 60%. Due to the alternating symmetrical cycles, it is found that the total true plastic deformation represents approximately 20% of the total value of the deformations. This phenomenon occurs due to the fact that the wood fibers stretch beyond their flow limit, which leads to the appearance of viscous plastic deformations. During the compression stress, a part of the deformations produced at traction is recovered, adding to these the shortening of the wood, after which the resumption cycle. Thus, the structural degradation of pine wood occurs by the loss of wood resilience to tensile-compression stresses, changes that can be observed qualitatively in Figure 9 and Figure 10, and quantitatively by changing mechanical properties. In practical applications, these phenomena can occur in the structural elements of the frames, at elements in the structure of stairs, furniture or buildings [24].

The microscopic structure of pine wood before loading can be seen in Figure 9, in the three main sections of wood. The annual rings are very distinctive and the differences between late wood and early wood are very visible (Figure 9a). Moreover, the resin canal can be observed in the transversal section. In the tangential-longitudinal section, the parenchyma rays with nuclei are very visible (Figure 9b). In the radial section, can be noticed the axially aligned tracheid cells with different sizes in late wood and early wood (Figure 9c). 

However, due to the anisotropic characteristics of wood and the different strength of late wood and early wood layers, the fracturing mechanism does not occur in one direction [4]. Therefore, the rupture of the pine wood samples occurred at an angle of approximately 30 degrees to the direction of the fiber, as shown in Figure 10d. Because the failure plane is not perpendicular to the applied load direction, the tensile strength consists of the total or effective stress, which has normal and shear stresses as components, as claimed by Christoforo et al. [4] who studied the same phenomenon, but on tropical wood species. Analyzing the fracture of the pine wood samples subjected to tensile - compression uniaxial parallel to the fibers, it was found that the direction of normal stresses does not coincide with the main elastic directions, in addition to normal strains appear shearing strains, as shown by the generalized law of Hooke. At compression parallel to the fibers, the structural changes begin with the increase of the stress level, and the instability of the fibers is observed by the appearance of some macroscopic kinks [6,21,25]. The yielding of a material represents the transition from the initial state of a volume element from an elastic material to a different and irreversible state [12,26,27]. In the case of the specimen made of material with predominantly ductile behavior that is subjected to traction, the rupture occurs in successive stages (Figure 10): The peripheral areas yield under the action of tangential stresses, and the central area yields under the action of normal stresses.

Degradation of mechanical properties of wood samples subjected to symmetrical cyclic stresses accompanied by microstructure degradation of wood. If the structural damage in tension does not affect the stiffness degradation or the strength of the wood when a reversed load is applied in compression (stiffness recovery) as was shown in Figure 4, in contrary, the constitutive damage does reduce the tensile capacity of the material when a reverse loading from compression to tension is applied as Sirumbal-Zapata et al. [27] demonstrate in their study. In Figure 10b–d, all effects of mechanical phenomena as tensile softening brittle failure, stiffness cyclic degradation, and recovery after load-reversal, as well as compressive permanent plastic deformation due to ductile failure can be observed at microscopic levels. Gong and Smith [1], Pinho et al. [12], and Gutkin at al. [26] explain the phenomenon of kinks by the fact that tracheids in late wood have a higher rigidity than those in early wood. Thus, the first kinks develop between the late wood tracheids, at the boundary between two growth rings, the phenomenon spreading to the early wood tracheids. After cyclic traction-compression stresses, the buckling of the tracheids in the early wood results. 

## 4. Conclusions

In this study, axially symmetrical cyclic loads (traction-compression) samples of pine wood in the direction parallel to the grain were tested. The degradation of the mechanical properties begins by the degradation of the wood structure as a result of the different resistance of the early wood and late wood layers, as well as the different viscous elastic responses to the traction and compression of the wood. Thus, after 20 alternately symmetrical cycles, the longitudinal modulus of elasticity decreases by approximately 28%, and with the increase of the loading speed, for some cases, there was even destruction of the samples. Also, the way of breaking the wood that occurs at an angle of approximately 30 degrees is noticed, which implies the appearance of shear stresses during the axially symmetrical cyclic stress. This study reveals that, in the strength calculations and in the design norms, it is necessary to introduce some coefficients that take into account the speed of wood degradation at cyclical stresses.

## Figures and Tables

**Figure 1 polymers-12-02176-f001:**
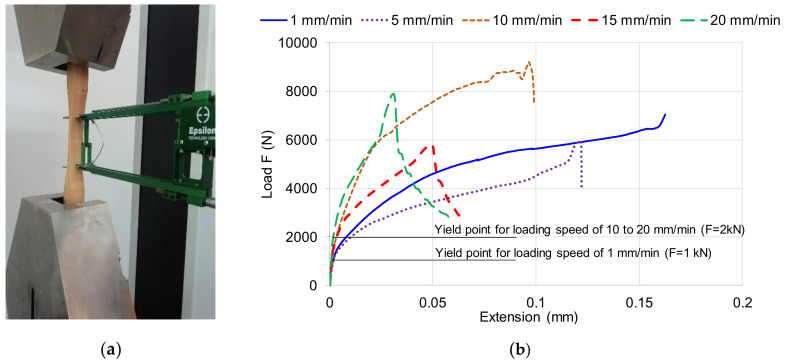
The preliminary test: **(a)** tensile test; **(b)** characteristic curves of pine wood subjected to tensile with different loading speeds.

**Figure 2 polymers-12-02176-f002:**
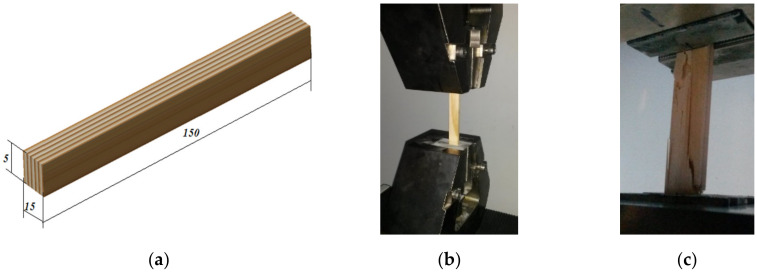
Experimental set-up: (**a**) the geometry of tested samples; (**b**) details of the sample in clamping jaws of the testing machine during the symmetric cyclic loading to tensile-compression; (**c**) Failure mode of sample tested to cyclic loading.

**Figure 3 polymers-12-02176-f003:**
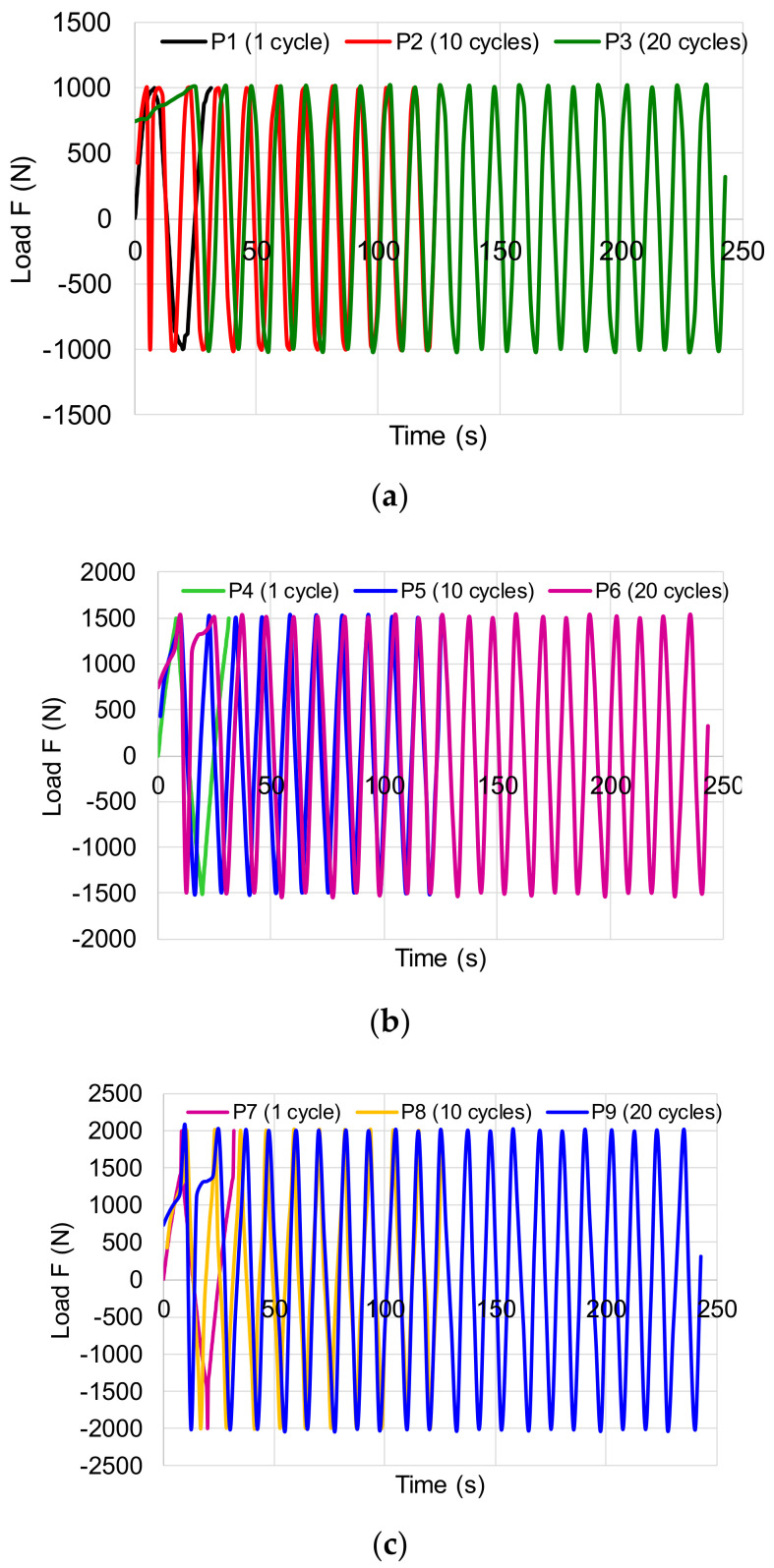
Variation of symmetric cyclic loading to tensile-compression: (**a**) ±1000 N; (**b**) ±1500 N; (**c**) ±2000 N.

**Figure 4 polymers-12-02176-f004:**
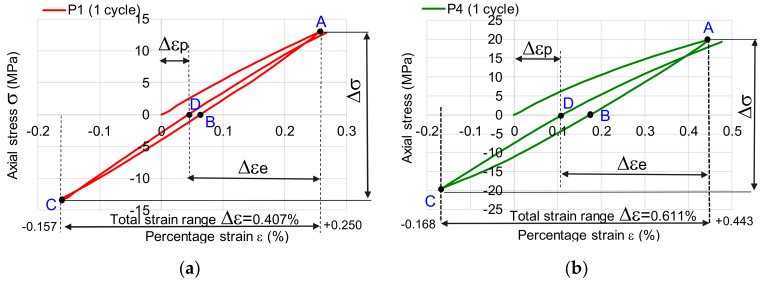
Hysteresis loops: (**a**) case of load speed of 1 mm/min and *F* = ± 1kN; (**b**) case of load speed of 10 mm/min and *F* = ± 1kN; (**c**) case of load speed of 1 mm/min and *F* = ± 1.5 kN; (**d**) case of load speed of 10 mm/min and *F* = ± 1.5 kN; (**e**) case of load speed of 1 mm/min and *F* = ± 2kN; (**f**) case of load speed of 10 mm/min and *F* = ± 2kN.

**Figure 5 polymers-12-02176-f005:**
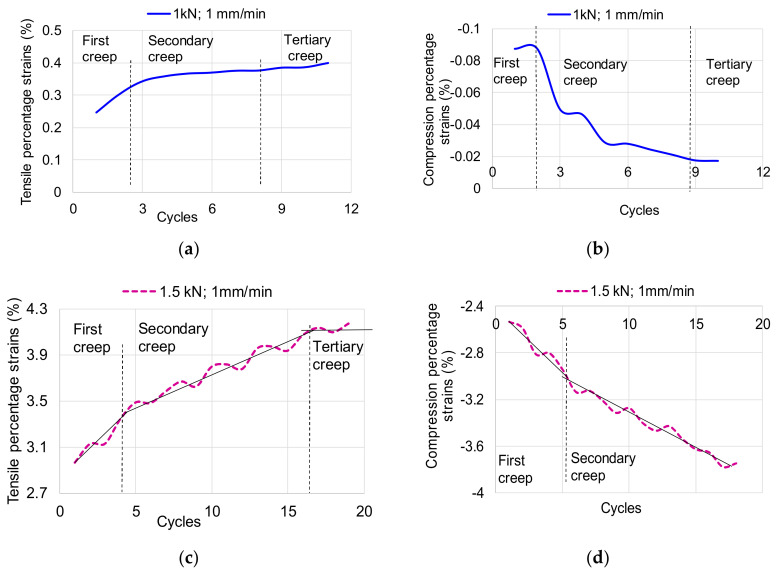
Master curves: (**a**) tensile strains of sample P3 subjected to 1 mm/min and *F* = ± 1kN; (**b**) compression strains of sample P3 subjected to 1 mm/min and *F* = ± 1kN; (**c**) tensile strains of sample P6 subjected to 1 mm/min and *F* = ± 1.5 kN; (**d**) compression strains of sample P6 subjected to 1 mm/min and *F* = ± 1.5 kN; (**e**) tensile strains of sample P9 subjected to 1 mm/min and *F* = ± 2kN; (**f**) compression strains of sample P9 subjected to 1 mm/min and *F* = ±2 kN.

**Figure 6 polymers-12-02176-f006:**
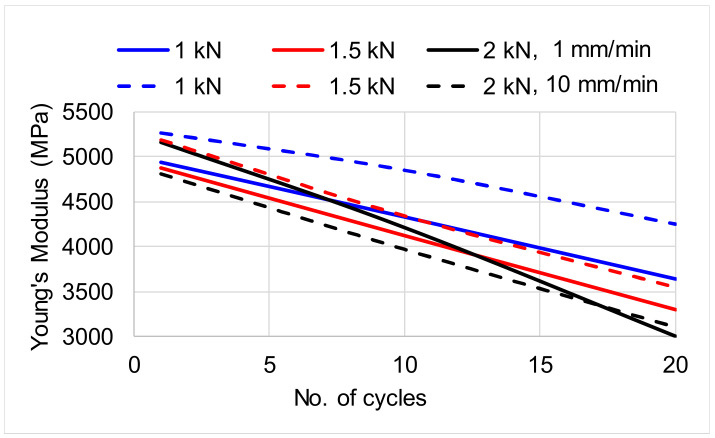
Degradation rate of Young’s Modulus with cyclic loading.

**Figure 7 polymers-12-02176-f007:**
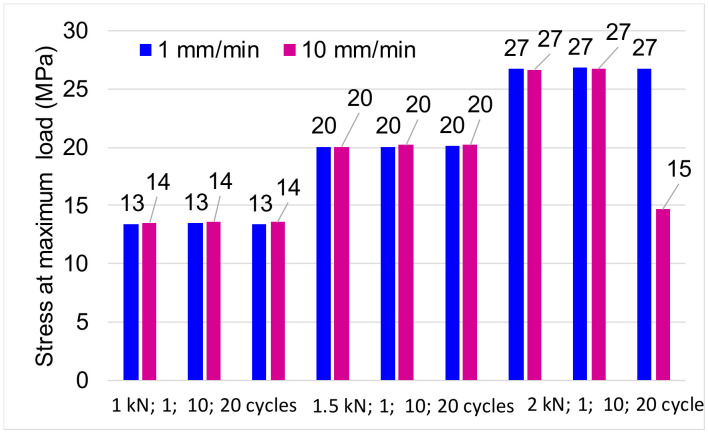
Variation of stresses at maximum load.

**Figure 8 polymers-12-02176-f008:**
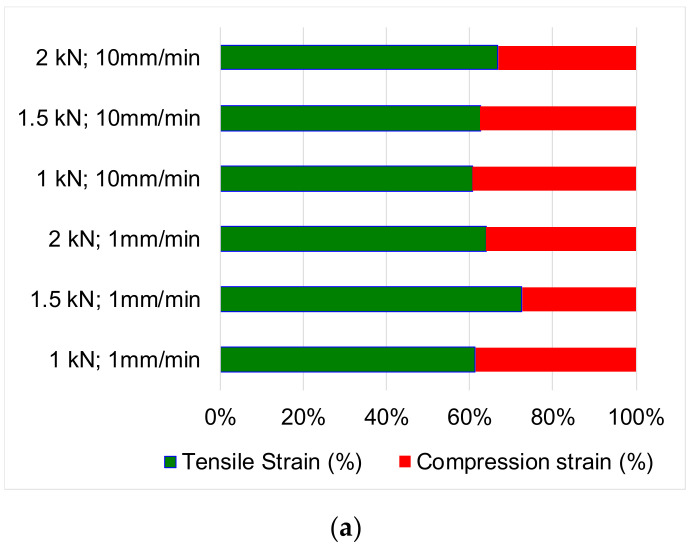
Comparison of strains in cases of different load speeds and intensity: (**a**) tensile and compression strains; (**b**) total true elastic and plastic strain range after a complete cycle.

**Figure 9 polymers-12-02176-f009:**
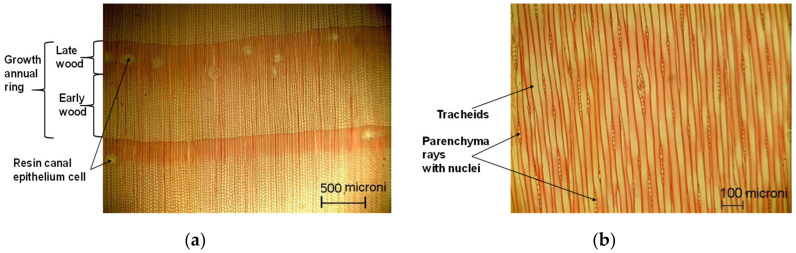
The microscopic view of Scots pine wood: (**a**) Cross section, magnification ×500; (**b**) longitudinal—tangential section, magnification ×100; (**c**) longitudinal—radial section, magnification ×100.

**Figure 10 polymers-12-02176-f010:**
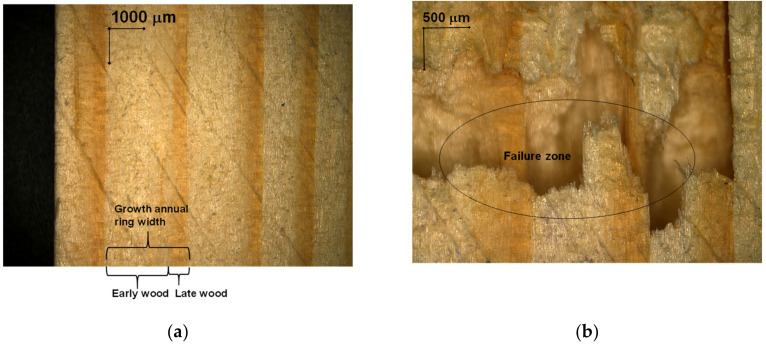
The structure of pine samples in different degradation stages: (**a**) initial macros-structure of wood samples, before test; (**b**) optical view of wood fracture after test; (**c**) microscopic view of tracheid cells failure, magnification ×100; (**d**) failure trend line, magnification ×100; (**e**) schematic representation of traction deformation; (**f**) schematic representation of wood failure due to anisotropy.

**Table 1 polymers-12-02176-t001:** Input data for experimental tests.

Set of Tests	Sample	No of Cycles	Gauge Length (mm)	Speed (mm/min)	Width (mm)	Thickness (mm)	Area (mm^2^)	Limit of Tensile/ Compression Loading (kN)
A1	P1	1	60	1	15	5	75	±1
	P2	10	60	1	15	5	75	±1
	P3	20	60	1	15	5	75	±1
B1	P4	1	60	1	5	5	75	±1.5
	P5	10	60	1	15	5	75	±1.5
	P6	20	60	1	15	5	75	±1.5
C1	P7	1	60	1	15	5	75	±2
	P8	10	60	1	15	5	75	±2
	P9	20	60	1	15	5	75	±2
A10	P10	1	60	10	15	5	75	±1
	P11	10	60	10	15	5	75	±1
	P12	20	60	10	15	5	75	±1
B10	P13	1	60	10	15	5	75	±1.5
	P14	10	60	10	15	5	75	±1.5
	P15	20	60	10	15	5	75	±1.5
C10	P16	1	60	10	15	5	75	±2
	P17	10	60	10	15	5	75	±2
	P18	20	60	10	15	5	75	±2

**Table 2 polymers-12-02176-t002:** The strains values collected for first cycle, where Δε is the total strain range; Δε**_e_**—the true total elastic strain range; Δε**_p_**—the true total plastic strain range; ε**_t_**–elongation strain; ε**_c_**—compression strain.

Set of Tests	Δε = ε_t_ +ε_c_ (%)	Δε_e_ (%)	Δε_p_ (%)	ε_t_ (%)	ε_c_ (%)
A1	0.407	0.2042	0.0458	0.250	0.157
B1	0.611	0.3422	0.1008	0.443	0.168
C1	0.969	0.5368	0.0832	0.620	0.349
A10	0.402	0.1982	0.0458	0.244	0.158
B10	0.639	0.3212	0.0788	0.400	0.239
C10	1.036	0.5566	0.1354	0.692	0.344

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
