# Peer review of "Degradation of Mechanical Properties of Pine Wood Under Symmetric Axial Cyclic Loading Parallel to Grain"

_polymers, 2020, doi:10.3390/polym12102176_

Round 1
Reviewer 1 Report
The paper “Degradation of Mechanical Properties of Pine Wood Under Symmetric Axial Cyclic Loading Parallel to Grain” aims to analyze the viscoelastic behavior of pine wood subjected to symmetric axial cyclic loading under different conditions of loads and loading speeds. The subject is novel and there are really significant contents of wood fatigue need to be investigated thoroughly. However, the presented work does not fully bring out the importance and beauty in some aspects.
- The abbreviations of MOR, DMA are not interpreted in Part 1 Introduction.
- In Part 2 Materials and Methods, it is necessary to specify what ASTM standard is utilized in the cyclic loading experiment, the static tensile experiment is also needed to determine the intrinsic mechanical property of wood samples.
- In Part 3 Results, there are some mistakes in Fig. 3 and Fig. 4 (b).
- The contents of Fig. 4 (the hysteresis loop for the first stress cycle) as well as Fig. 5 (the variations of the successive displacements at all the tensile stresses cycles) are not illustrated thoroughly, besides the phenomenal description. What is the purpose of the curve fitting in Fig.5?
- It is better to mark the annual rings, the parenchymatic rays and the axially aligned tracheid cells in Fig. 9. A more detailed discussion is needed for Fig. 9 and Fig. 10, respectively.
- According to the fatigue experimental tests, the three creep regions specific to wood was observed, the degradation of mechanical properties and structural failure were obtained. However, the viscoelastic behavior of pine wood under symmetric axial cyclic loading is not investigated enough.
Author Response
We are thankful to reviewer for all useful recommendation which makes the paper more understandable. We corrected the language and improved the manuscript. The paper was rewritten taking into account the suggestions. The manuscript was extensively modified.
Please find attached the Authors' reply to Reviewer 1.

Reviewer 2 Report
please see my comments in the file attached

Author Response
We are thankful to reviewer for all useful recommendation which makes the paper more understandable. We corrected the language and improved the manuscript. The paper was rewritten taking into account the suggestions. The manuscript was extensively modified.
In attached file, we send the response to reviewer 2.

Reviewer 3 Report
In my opinion, this is a well-structured and written document, pleasant to read and of obvious interest from both fundamental and applied points of view for the scientific community on wood. This paper aims at analyzing the viscoelastic of pinewood, distinguishing the three creep regions specific to wood, and highlighting the degradation of its longitudinal elastic properties at high number of tensile-compression cycles.
I just recommend re-reading the document to eliminate the few typos that appear throughout the document and adding few information:
- In abstract: “three creep regions” (write in plural)
In introduction: “The tension” (capital letter at the beginning of sentence)
In materials’ section 2.1: “(sample P2)”, “(sample P3)” (replace “test” by “sample” and add parentheses)
In results’ section 3: “(Figures 5, a; c and e)” (add comma and semicolon); “Figures 5, b; d and f” (write in plural); “late wood” (add space); “(Figure 9, b)” (capital letter)
- What does mean “kink initiation”? “kink growth”? Is it possible to specify?
- Please add a column for group numbering in Table 1 (A1, B1, C1, A10, B10 and C10) in accordance with the text.
Author Response
We are thankful to reviewer for all useful recommendation which makes the paper more understandable. We corrected the language and improved the manuscript. The paper was rewritten taking into account the suggestions. The manuscript was extensively modified.
In attached file, we send the response to reviewer 3

Reviewer 4 Report
Manuscript polymers-926207, "Degradation of mechanical properties of pine wood under symmetric axial cyclic loading parallel to grain" proposes an interesting experimental study of mechanical behaviour of Scots pine wood and the corresponding morphological changes, when submitted to symmetric axial cyclic loading in grain direction.
The experimental work is correctly carried out, but the discussion should be improved.
All in all, I will recommend for publication the revised manuscript, only after solving/correcting the different important aspects mentioned in my comments.

Author Response
We are thankful to reviewer for all useful recommendation which makes the paper more understandable. We corrected the language and improved the manuscript. The paper was rewritten taking into account the suggestions. The manuscript was extensively modified.
In attached file, we send the response to reviewer.

Round 2
Reviewer 1 Report
The manuscript has been improved greatly. I accept it in present form.